# Instance-Based Learning Following Physician Reasoning for Assistance during Medical Consultation

Matías Galnares *, Sergio Nesmachnow * and Franco Simini *

Universidad de la República, Montevideo 11200, Uruguay

\* Correspondence: matias.galnares@fing.edu.uy (M.G.); sergion@fing.edu.uy (S.N.); simini@fing.edu.uy (F.S.)

**Abstract:** This article presents an automatic system for modeling clinical knowledge to follow a physician's reasoning in medical consultation. Instance-based learning is applied to provide suggestions when recording electronic medical records. The system was validated on a real case study involving advanced medical students. The proposed system is accurate and efficient: $2.5\times$ more efficient than a baseline empirical tool for suggestions and two orders of magnitude faster than a Bayesian learning method, when processing a testbed of 250 clinical case types. The research provides a framework to implement a real-time system to assist physicians during medical consultations.

**Keywords:** computational intelligence; medical assistance; instance-based learning; healthcare; clinical decision support systems





## 1. Introduction

The search for better medical practices is a perpetual challenge for modern medicine. In this regard, computational intelligence has emerged as a promising subject for developing smart systems in healthcare practice [1]. Computational intelligence allows implementing automatic tools, enabling physicians to provide patients with a better quality of attention by performing early and accurate diagnosis and improving treatment. Furthermore, automatic systems and technologies based on computational intelligence have proven to be useful solutions to be applied in clinical practice. Some important advantages of intelligent automatic methods over traditional ones include better efficiency, accuracy, consistency, more time available for face-to-face consultation, and more time for critical tasks and critical cases, among others [2].

A specific subject where the learning capabilities of computational intelligence methods is very helpful to improve medical practice is analyzing and processing electronic medical records (EMRs). EMRs refer to digital records, collected by the individual medical practice, that contain the general health information of patients [3]. They usually consist of several types of health data, including, but not limited to, demographics, medical family history, medication, allergies, test results, and radiology images.

Currently, the majority of medical history recording products are based on predefined templates, which provide very limited freedom for writing patients medical records. Structured data entry is a hindrance to the usability of medical record applications, and is frowned upon by physicians who usually prefer to document using free text [4]. In addition, structured data entry systems do not take into account the particularities of the annotations of each physician, failing to effectively record the singularities of medical consultations. Alerts and suggestions offered by conventional products are generally based on previously defined rules, or according to mechanisms whose behavior remains the same throughout its operational life. The dissatisfaction of physicians with actual medical history recording products is increasing as they gain knowledge about automatic assistant tools. Consequently, physicians are increasingly aspiring to have sophisticated tools that help facilitate their clinical practice during medical consultations.

The research reported in this article is motivated by the need to further explore new ways of capturing, storing, and fostering medical reasoning. Thus, a formal proposal must be conceived to provide an accurate tool capable of following medical reasoning, aiming at helping physicians during medical consultations. In this line of work, this article presents a novel approach to represent clinical knowledge, which supports an appropriate methodology to follow reasoning in medical consultation. Likewise, the proposed representation does not pose formal restrictions to physicians, as they usually find when using common clinical data entry systems. An instance-based learning method is also introduced to provide suggestions in order to help during the process of registering a medical consultation. The developed system extends Praxis [5], a software used to follow medical reasoning with no templates, based on the accumulation of case types used to provide suggestions for subsequent cases.

The proposed approach was evaluated for a case study in which more than 50 advanced medical students had collaborated. Students tested the feasibility of the approach by using a proof-of-concept prototype. The performance of the proposed learning method was found to be satisfactory after being evaluated on 250 real instances constructed by the students. Results showed that the learning method was able to produce suggestions in a reasonable time frame, even when processing large volumes of data. The results suggest that the proposed approach was useful to accelerate the process of taking notes, since a convergence towards a high speed of completed medical records was observed. A high potential impact on clinical care may be projected, considering that the results showed that the proposed approach was appropriate to follow physician reasoning, especially during medical consultations. As a benchmark, 62% of the students were able to speed up writing time during medical consultations.

The main contributions of the research reported in this article include: (i) a formal structure to accurately represent clinical knowledge, and support the main flows of medical consultations; (ii) an instance-based learning method able to help reduce the time it takes to write notes; and (iii) a novel tool to help meet healthcare goals, which reminds physicians to record essential data to fulfilling care goals.

The article is structured as follows. Section 2 introduces learning models for assistance in medical consultation. A review of related work on learning models for assisting medical professionals is presented in Section 3. Section 4 describes a model proposed for representing clinical knowledge and patient history. Several flows to address relevant scenarios of medical consultations are presented in Section 5. The main implementation details of the proposed instance-based learning method are described in Section 6. Sample results from the evaluation are presented in Section 7. Section 8 discusses the usability of the proposed method and main strategies to improve the results and reduce uncertainties. Finally, Section 9 presents the main conclusions of the research.

## 2. Learning Models for Assistance in Medical Consultation

Despite the fact that physicians are becoming increasingly familiar with electronic medical records, they continue to have difficulties in dealing with long lists of pre-conceived variables, usually included in EMR systems. Although conventional EMR systems are useful to achieve legible, accessible, and complete documentation of medical consultations, they are causing several difficulties for physicians who adopt them. In many cases, physicians spend a lot of time searching for an option that allows them to record what they really want to write. Unfortunately, conventional EMR systems are template-based products that generate poor quality data, due to long search mechanisms and excessive mandatory fields, which often add noise to the relevant patient information [4]. Worse, the time required to enter clinical information sometimes exceeds the time required to write it on paper. The rigid structure of the templates to be filled-in during medical consultations does not fit the reasoning of physicians, nor their way of thinking.

Improvements in medical consultation assistance could be achieved by taking advantage of systems that allow better management of clinical information. To achieve better

assistance, physicians should be provided with new healthcare tools, considering that healthcare assistance during medical consultations is improved when the physician is able to:

(i)　Efficiently record all the information of a medical consultation, by reducing the time spent on mere data entry in order to gain more time to interact with the patient.
(ii)　Use automatic clinical suggestions to reach an accurate diagnostics, or an appropriate indication of treatments.
(iii)　Reduce medical errors, resulting from the human condition of the professional.
(iv)　Record each medical consultation considering the special relevance of the interoperability of clinical information.
(v)　Reuse recorded information for statistical and research purposes.

Computational intelligence can be applied to solve the deficiencies of current EMRs. Machine learning methods can be used to learn features from previous registered healthcare data sets, in order to provide suggestions for diagnoses and treatments based on information previously registered. By applying computational intelligence, systems can automatically identify solutions of similar clinical cases and can subsequently incorporate the knowledge gained to assist physicians during medical consultations. Learning methods can also contribute to reduce error-prone steps during the sequence of clinical tasks and decisions. Inevitable errors of human-based clinical practice may be reduced, such as drug contraindications, medication allergies, adverse drug reactions, and forgetting recurrent aspects of chronic patients. Furthermore, machine learning methods can progressively enhance their accuracy based on feedback provided by their own use.

An effective medical informatics support system must be adapted to the real health environment. In addition, a clinical evaluation of the usefulness of the system in real clinical work should be considered to determine its real capacity during clinical practice.

### 3. Related Works

This section reviews related works regarding learning models that assist medical professionals during their clinical activities.

Decision support systems can detect patterns, provide recommendations, and predict future behaviors for clinical practice. Wang et al. [6] proposed an Intelligent Self-Learning EMR (ISLEMR) system used to generate treatment recommendations based on learning and patient similarity. ISLEMR considers a group of ad hoc similarity metrics, considering patient diagnoses, demographic data, vital signs, structured lab test results, and information from external systems. The patient information is used to present an ordered menu with inferred recommendations for treatment plans. The system was validated on a real case study in Beijing, China in 2014, considering data from twelve-thousand patients. Precision results up to 80% were achieved for the first 10 items of the recommended menu; however, the applied learning algorithm only considered structured data, which implies less precision in determining similarities of clinical cases. Klann et al. [7] proposed a learning approach based on Bayesian networks (BN) to generate adaptive and context-specific treatment menus from past clinical information of patients. Each menu recommends a starting point for physicians, suggesting an initial draft to treat a specific situation. The BN models the probabilistic relationships among orders and diagnoses, covering typical scenarios from different aspects of medicine. The system was evaluated on a hospital simulation, demonstrating accurate predictive capabilities and outperforming a similar association rule mining approach, especially over less frequent cases. Support vector machines (SVM) have also been applied as learning models for medical assistance. Nakai et al. [8] applied SVM to predict clinical practices to be prescribed by using the information from previous practices of the same patient. The validation over real data from the Japanese system for medical billing proved the high precision of the model when facing frequent clinical cases; however, low precision results were obtained when dealing with less common cases. Barbantan et al. [9] proposed a medical decision support system using SVM and natural language processing to discover relations between medical concepts.

The model was successfully used to identify relations between medical concepts to help diagnoses, medication predictions, and to detect health patterns in Boston, USA. Shen et al. [10] proposed a multi-agent case-based reasoning approach for clinical decisions. The system searches clinical cases by identifying important words and terminologies, whereas medication allergies, adverse drug reactions, coexisting diseases, and other complications are evaluated to discard candidate cases. The system achieved a 78% matching rate for illnesses with simple syndromes. Installé et al. [11] developed a clinical data miner software framework for supporting clinical diagnostic using electronic case report forms (eCRF) based on templates and spreadsheets. Machine learning techniques are applied over the information gathered by the eCRF. A survey indicated that the system was considered user-friendly, and all physicians approved the possibility of using it in their own future works. Zieba [12] proposed a service-oriented support decision system for the diagnose of medical problems using web services with learning capabilities applying SVM. The system was evaluated using ontological datasets and it was able to predict a diagnosis by generating decision rules with acceptable accuracy values. Benmimoune et al. [13] designed a hybrid medical platform to assist physicians during their clinical reasoning process using rule-based reasoning (RBR) for general clinical cases and case-based reasoning (CBR) for clinical experiences. The proposed platform gathers relevant information about the patient status using an adaptive questionnaire and searches for the most similar stored case, following the CBR approach. If no similar case is found, the platform applies an RBR approach to deduce a solution according to rules defined by medical experts. Neither the implementation nor the prototype of the proposed system was described. Wilk et al. [14] proposed a framework to assist patients with multi-morbidity conditions, considering patient preferences for suggesting customized clinical practice guidelines. Clinical guidelines are modeled using actionable graphs and first-order logic, and a secondary medical knowledge component is used to identify adverse interactions resulting from conflicting therapies. A high-level proof of concept implementation was presented to show the feasibility of the proposed framework but no real evaluation was proposed.

Praxis is an electronic medical records application, developed to streamline the entry of clinical data and improve medical practice [5]. It emulates the processes that physicians follow when they are recording clinical information. The software uses previously entered information to offer recommendations for registering a new consultation, according to the past practice of the physician user (i.e., suggesting a set of cases similar to the one being evaluated). Praxis applies an empirical approach and has been gradually improved over more than twenty-five years, to fit the North American medical system. Praxis does not apply computational intelligence to build an expert system for the recommendation of diagnoses and treatments.

A summary of related works reviewed in this section is presented in Table 1, reporting for each article the methods applied, the most relevant features of each research, and any identified weaknesses.

The analysis of related works allowed identifying several proposals applying computational intelligence and other learning-based methods for diverse health scenarios. Most existing systems focuses on providing suggestions for treatments and diagnoses, based on similarity metrics regarding relevant information from past medical assistance. Reviewed works are able to identify similar clinical cases in order to provide suggestions for diagnoses, prognosis, and treatments. Furthermore, they contribute to reducing error-prone steps during the clinical process. The system presented in this article contributes to this line of research, including specific differences with existing related works: it supports non-structured free text information to be used in the learning process, instead of just structured information [6]; a more effective learning approach is applied, which outperforms a Bayesian learning method such as the ones that have been previously used in the related literature [7]; suggestions are generated considering all similar case types (of different patients), instead of just previous information of the same patient [8]; and it does not rely

on complex rules based on natural language processing, which limits the applicability of other suggestion systems [9].

**Table 1.** Summary of reviewed works.

| Work | Method(s) | Relevant Features | Weaknesses |
|---|---|---|---|
| Wang et al. [6] | Ad hoc patient similarity algorithm. | Menu of inferred recommendations, real-time feedback. | Only considers structured data. |
| Klann et al. [7] | Bayesian networks. | Suggest initial drafts, reduce workload of physicians. | Relies on a small set of orders and diagnoses. |
| Nakai et al. [8] | Linear support vector machine. | Use information from previous practices, high precision for common cases. | Low precision when dealing with less common cases. |
| Barbantan et al. [9] | Natural language processing, support vector machine classifier. | Medical structured-related concept model, detect patterns about patient health. | Only evaluated on clinical phrases with more than one medical concept. |
| Shen et al. [10] | Language analysis, ad hoc matching. | Suggest diagnoses, prognosis and treatments. | Knowledge representation fails to analyze evolutionary contexts. |
| Installé et al. [11] | Preprocessing, machine learning techniques. | Reduce error-prone steps during diagnostics, user-friendliness. | Variable length array types not supported, not useful for longitudinal data capture. |
| Zieba [12] | Cost-sensitive support vector machine. | Web services with learning capabilities, generate decision rules. | Only acceptable accuracy values of decision rules. |
| Benmimoune et al. [13] | Rules for generic cases, case-based reasoning component. | Adaptive questionnaire according to patient profile. | No prototype was implemented. |
| Wilk et al. [14] | Actionable graphs. first-order logic. | Clinical guidelines for multi-morbidity conditions, considers patient preferences. | No real evaluation. |

## 4. Clinical Knowledge Model to Follow Physician Reasoning

A formal model is proposed for representing clinical knowledge and patient history, including medical records.

### 4.1. Clinical Knowledge Base

A bottom-up modeling approach is used to present the proposed clinical knowledge model. Several entities are defined in order to specify a clinical knowledge base that describes information of real medical case types. All entities included in a clinical knowledge base are described in the following subsections.

#### 4.1.1. Unit of Thought

As defined by Low [15], a unit of thought is a statement that describes a basic clinical idea. Let $UT^M$ be a unit of thought registered by physician $M$. $UT^M$ is denoted as $UT^M = <ptext, uqcn, uqpt, exph, terms, inuse, ctSchedule, M>$, where *ptext* denotes a string capable of containing structured or random data, *uqcn* indicates if the unit refers to information to be used only in a unique consultation, *uqpt* indicates if the unit refers to unique information of a specific patient, *exph* indicates if the unit contains exclusive data for physician use, *terms* detail associations with health terminological standards, *inuse* denotes if the unit is in use during a consultation, and *ctSchedule* indicates the frequency that a unit appears in a case type. A unit of thought used in a case type will reappear each time the case type is used, unless a specific frequency is defined by its *ctSchedule* attribute.

The set of all units of thought registered by physician $M$ is denoted as $UT_T^M$. Let $UT_1^M$ = $<ptext_1, uqcn_1, uqpt_1, exph_1, terms_1, inuse_1, ctSchedule_1, M>$ and $UT_2^M$ = $<ptext_2, uqcn_2, uqpt_2, exph_2, terms_2, inuse_2, ctSchedule_2, M>$ be units of thought registered by physician $M$. A constraint on units of thought is defined in Equation (1), implying that each basic clinical idea is represented by a unique unit of thought.

Considering that text variations do not change the meaning of a basic clinical idea, an ad hoc function *equal* (defined in Equation (2)) is necessary to identify if two phrases represent the same clinical idea. The same clinical idea can be instanced containing both structured information and random data, which implies that two different text strings can represent the same clinical idea.

$$
\left.\begin{array}{c}
UT_1^M \in UT_T^M \\
UT_2^M \in UT_T^M \\
equal(ptext_1, ptext_2)
\end{array}\right\} \Rightarrow UT_1^M = UT_2^M
\tag{1}
$$

$$
equal(t_1, t_2) = \begin{cases} true & \text{if } t_1 \text{ and } t_2 \text{ describe} \\ & \text{the same clinical idea.} \\ false, & \text{otherwise.} \end{cases}
\tag{2}
$$

### 4.1.2. Conceptual Element

A conceptual element is composed of a set of units of thought grouped to represent a broader concept. Several attributes are used to model all possible features of a conceptual element. Let $CE^M$ be a conceptual element registered by physician $M$. $CE^M$ is denoted as $CE^M = <name, display, chron, setDesc>$, where *name* denotes the name of the element, *display* indicates the default display mode of its units of thought, *chron* indicates if the element refers to a chronic condition, and *setDesc* denotes a set of possible descriptors of the conceptual element. The set $setDesc = \{[desc_1, subset_1(UT_T^M)], ..., [desc_k, subset_k(UT_T^M)]\}$ is composed of several pairs, each one is used to model a possible option to describe a real condition of a conceptual element.

Two constraints are defined on conceptual elements. The constraint presented in Equation (3) implies that a conceptual element is identified by its *name*.

$$
\left.\begin{array}{c}
CE_1^M = < name_1, display_1, chron_1, setDesc_1 > \\
CE_2^M = < name_2, display_2, chron_2, setDesc_2 > \\
name_1 = name_2
\end{array}\right\} \Rightarrow CE_1^M = CE_2^M
\tag{3}
$$

The constraint presented in Equation (4) implies the uniqueness of each descriptor into a conceptual element. Several units of thought can be labeled under the same descriptor to define an identified sub set, describing a real condition of an element.

$$
\left.\begin{array}{c}
[desc_1, subset_1(UT_T^M)] \in setDesc \\
[desc_2, subset_2(UT_T^M)] \in setDesc \\
desc_1 = desc_2
\end{array}\right\} \Rightarrow subset_1(UT_T^M) = subset_2(UT_T^M)
\tag{4}
$$

### 4.1.3. Conceptual Component

A conceptual component is composed of a set of conceptual element references, grouped to define sections of clinical information. Each conceptual component represents a typical clinical data section, in which a physician generally groups the information of a medical consultation.

Let $CC^M = <id, secType, activeElems>$ be a conceptual component defined by physician $M$, identified by its *id* attribute. The *secType* attribute is used to represent the type of data section, such as physical examination, medicines, and laboratory indications. Each *secType* must belong to the *ALL-SECTION-TYPES* set, which models all possible sections of the

patient medical records. The set *activeElems = {[elemName$_1$, activeDesc$_1$], ..., [elemName$_k$, activeDesc$_k$]}* indicates which descriptor is used for each conceptual element referenced in a conceptual component.

Two constraints are defined on the conceptual components domain. The constraint presented in Equation (5) implies that a conceptual component is identified by its *id* attribute.

$$
\left.
\begin{array}{l}
CC_1^M = < id_1, secType_1, activeElems_1 > \\
CC_2^M = < id_2, secType_2, activeElems_2 > \\
id_1 = id_2
\end{array}
\right\} \Rightarrow CC_1^M = CC_2^M
\tag{5}
$$

A second constraint presented in Equation (6) ensures the referential integrity of names and descriptors of the active elements, referenced from a conceptual component.

$$
\left.
\begin{array}{l}
CC^M = < id, secType, activeElems > \\
[elemName, activeDesc] \in activeElems
\end{array}
\right\} \Rightarrow
\begin{array}{l}
\exists \text{ conceptual element } e = < name, ..., setDesc > / \\
e.name = elemName \wedge \exists \text{ d} \in setDesc, \\
d.desc = activeDesc
\end{array}
\tag{6}
$$

### 4.1.4. Case Type

Several conceptual components can be grouped by a unique name to label a complex scenario, representing a real case type that can occur during a physician's workday. Let $CT^M$ be a case type registered by physician $M$. $CT^M$ is denoted as $CT^M = <name, \{CC_1^M, ..., CC_n^M\}, chron, chronicComponents>$, where *name* indicates the name of the case type, the set $\{CC_1^M, ..., CC_n^M\}$ describes a specific group of conceptual components, *chron* indicates if the case type is marked as chronic, and *chronComponents* denotes all components used to monitor chronic conditions.

Three constraints are defined on case types domain. The constraint presented in Equation (7) implies that a case type is identified by its *name*.

$$
\left.
\begin{array}{l}
CT_1^M = < name_1, chron_1, comps_1, chronComponents_1 > \\
CT_2^M = < name_2, chron_2, comps_2, chronComponents_2 > \\
name_1 = name_2
\end{array}
\right\} \Rightarrow CT_1^M = CT_2^M
\tag{7}
$$

The second constraint presented in Equation (8) implies that each conceptual component of a case type models a different section of the clinical information.

$$
\left.
\begin{array}{l}
CT^M = < nc, \{CC_1^M, ..., CC_n^M\}, chron, chComps > \\
CC_i^M = < id_i, secType_i, subset_i) > \\
CC_j^M = < id_j, secType_j, subset_j) >
\end{array}
\right\} \Rightarrow
\begin{array}{c}
secType_i = secType_j \\
\Updownarrow \\
i = j \ \forall i, j \in \{1, n\}
\end{array}
\tag{8}
$$

The third constraint presented in Equation (9) implies that each chronic conceptual component models a different section of chronic clinical information.

$$
\left.
\begin{array}{l}
CT^M = < nc, comps, true, \{CC_{chron_1}^M, ..., CC_{chron_m}^M\} > \\
CC_{chron_i}^M = < id_i, secType_i, subset_i) > \\
CC_{chron_j}^M = < id_j, secType_j, subset_j) >
\end{array}
\right\} \Rightarrow
\begin{array}{c}
secType_i = secType_j \\
\Updownarrow \\
i = j \ \forall i, j \in \{1, m\}
\end{array}
\tag{9}
$$

Finally, the clinical knowledge base (CKB) of a physician $M$ is defined as $CKB^M = \bigcup_{i=1}^{n} CT_i^M$. i.e., the union of all case types registered by physician $M$.

### 4.2. Patient Representation

A data structure is used to organize the information of each patient, considering the most relevant groups of personal data sets. The proposed structure includes medical records of a patient's history, and it also considers the chronic information of each patient.

#### 4.2.1. Patient Structure

Each patient is modeled as $P = <personalData, MR^P, chronicElems, chronicCaseTypes>$ where *personalData* denotes personal data (sush as patient and family background), $MR^P$ denotes all medical records of the patient $P$, *chronicElems* indicates associations with chronic conceptual elements, and *chronicCaseTypes* indicates associations with chronic case types. The *chronicElems* set is defined as *chronicElems* = $\{[elemName_1, chronDesc_1], ..., [elemName_j, chronDesc_j]\}$, and it is used to remember the descriptors of the elements that describe the chronic conditions of a patient. Additionally, the set *chronicCaseTypes* = $\{caseTypeName_1, ..., caseTypeName_k\}$ is used to remember all chronic case types associated with a specific patient $P$.

#### 4.2.2. Patient Medical Records

The set of medical records of a patient $P$ is denoted by $MR^P$ and contains all records included in the medical history of the patient. A medical record of patient $P$ created at time $t$ is denoted as $mr_t^P$ and it is defined as $mr_t^P = <content,p,t>$, where *content* is a set of *[phrase, unit]* pairs, each one includes a unit of thought associated with a clinical phrase. Consequently, $MR^P = \{mr_{t_1}^P, mr_{t_2}^P, mr_{t_k}^P\}$ describes the history of a patient, containing $k$ medical records.

Let $mr_t^P = <content,p,t>$ be a specific patient medical record, where *content* = $\{[phrase_1, unit_1], ..., [phrase_n, unit_n]\}$ is composed by one or more pairs of clinical information. A function *showRecord* is used to print the content of a medical record, taking into account all phrases included in the *content* of a medical record. Function *showRecord* only prints clinical phrases, no unit of thought is shown.

#### 4.2.3. New Medical Record

Let $CKB_t^M = \{CT_1^M, CT_2^M, ..., CT_n^M\}$ be the composition of the clinical knowledge base of physician $M$ at time $t$. A medical record $mr_t^P$ is generated as a result of the interaction of physician $M$ and patient $P$, during a consultation at time $t$.

Since a physician usually takes a case type $CT_x^M$ as basis to record a specific consultation, a transformation $T^*$ can be applied to generate a new medical record. Consequently, a record $mr_t^P = T^*(CT_x^M)$ is created, taking into account the active information of a selected case type. The active information of a case type is defined by the units of thought with *inuse* attribute in true. Transformation $T^*: CKB^M \rightarrow MR^P$ is defined as $T^*(CT) = mr$, where $mr$ is generated by applying Algorithm 1.

---

**Algorithm 1** New medical record of patient $P$

---

1: units ← getAllUnitsIncludedIn(CT)

2: content ← ∅

3: **for** unit **in** units **do**

4:     **if** unit.inuse **then**

5:         **if** not (unit.uqpt or unit.exph) **then**

6:             itemCont ← [copyCurrentText(unit.ptext), unit]

7:             content ← content ⋃ {itemCont}

8:         **end if**

9:     **end if**

10: **end for**

11: $mr_t^P$ ← <content,P,t>

---

Algorithm 1 starts by getting all units of thought referenced in a case type *CT* (line 1). The algorithm iterates over all referenced units to identify units of thought marked with *inuse* attribute (lines 3–4). Further, units marked with *uqpt* or *exph* attributes are not taken into account for creating a new medical record (line 5). A new data pair is created for each identified unit (line 6), each pair includes the identified unit of thought, and a copy of its current text presentation. All new pairs are joined to build the full content of the consultation record (line 7). Finally, $mr_t^P$ is created as a new medical record, containing the full description of the consultation of patient *P* at time *t*.

## 5. Medical Consultation Flows

Different flows for address the most relevant scenarios that arise during medical consultations are presented. These scenarios describe usual situations of physician workday, including multiple diagnoses, and the attention of chronic patients.

### 5.1. Starting Attention of a Patient

Algorithm 2 details the first steps which occur during a medical consultations.

---

**Algorithm 2** Start attention of patient $P$

---

1: showPersonalInfo(P.personalData)

2: showChronicElementDescriptors(P.chronicElems)

3: chronicCTs ← getCaseTypesByNames(P.chronicCaseTypes)

4: **if** chronicCTs ≠ ∅ **then**

5:     All case types included in chronicCTs are suggested to physician

6:     Physician select $CT_{chron_1}^M$, ..., $CT_{chron_k}^M$ to be used as basis

7:     $CT_{merge}^M$ is build by merging $CT_{chron_1}^M$, ..., $CT_{chron_k}^M$ (Algorithm 8)

8:     applyCaseType(P, $CT_{merge}^M$) is called  (Algorithm 3)

9: **end if**

10: Show message agents according its trigger conditions

11: Physician continues with patient attention

---

The physician starts the attention of patient *P* by opening a registry editor to record the information of the new medical consultation. Personal information is loaded (line 1) to introduce the patient. All descriptors of chronic elements (line 2) and all chronic case types (line 3) associated with the patient are presented and suggested to the physician, who can select the chronic case types that are appropriate to being applied into the consultation

(lines 4–9). Before the physician continues with patient attention, all message agents are evaluated and shown according to its trigger conditions (line 10).

### 5.2. Selecting an Already Defined Case Type

The selection of an already defined case type allows the physician to efficiently reuse previously registered information. Algorithm 3 details how to apply a case type during a medical consultation.

Algorithm 3 starts by evaluating the *chronic* attribute of a case type $CT^M$ (lines 1–2). If the case type is identified as chronic, a specific method for applying a chronic case is called (line 3). Otherwise, all elements referenced in the *components* attribute are determined, and its units of thought are marked as in use according *setUnitsInUse* auxiliary procedure. The auxiliary procedure encapsulates the logic of how units of thought are activated. The algorithm continues by showing all units marked as in use, and highlighting the units that are exclusive for physician use (lines 8–9). Finally, each message agent that has $CT^M$ as a trigger condition is presented to the physician (line 10).

Procedure *setUnitsInUse* iterates over all conceptual elements of a case type (line 12). All units included in each conceptual element are identified (line 13), and each unit of thought is marked as in use according the values of its attributes (lines 14–25).

---

**Algorithm 3** applyCaseType($P, CT^M$)

---

1: $CT^M$ = <name, components, chronic, chronicComponents> is selected

2: **if** chronic **then**

3:     applyChronicCaseType(P, $CT^M$) (Algorithm 4)

4: **else**

5:     elements ← getAllElementsIncludedIn(components)

6:     setUnitsInUse(elements,$CT^M$)

7: **end if**

8: Show all units with *isuse* attribute in true

9: Highlight all units with *exph* attribute in true

10: Show message agents that have $CT^M$ as a trigger condition

11: procedure **setUnitsInUse**(elements,$CT^M$)

12: **for** element **in** elements **do**

13:     units ← getAllUnitsIncludedIn(elements)

14:     **for** unit **in** units **do**

15:         **switch** ()

16:         **case** unit.exph**:**

17:             unit.inuse = true

18:         **case** isTime(unit.ctSchedule, $CT^M$)**:**

19:             unit.inuse = true

20:         **case** element.display ∧ isEmpty(unit.ctSchedule)**:**

21:             unit.inuse = true

22:         **case** otherwise**:**

23:             unit.inuse = false

24:         **end switch**

25:     **end for**

26: **end for**

---

### 5.3. Chronic Patients Flow

A case type $CT^M$ can be marked as a chronic case type $CT^M_{chron}$ at any time. When a $CT^M_{chron}$ is marked as chronic, its *chron* attribute is activated and its *chronicComponents* attribute is initialized with an empty set. The chronic components are defined the first time that the case type is used to monitor a chronic patient. Algorithm 4 details how a physician can apply a chronic case type $CT^M_{chron}$.

Algorithm 4 analyzes if it is the first time that a chronic case type $CT^M_{chron}$ is used with a patient being evaluated (lines 1–2). In that case, elements referenced in usual conceptual components are determined, and its units of thought are activated by calling *setUnitsInUse* procedure (lines 3–4). If $CT^M_{chron}$ was used in any previous consultation of the same patient (line 6), then its *chronic* components are taken into account each time the physician decides to apply the case type, since *chronic* components are used to monitor the evolution of a chronic condition. However, the first time that $CT^M_{chron}$ is used to monitor the evolution of a patient, the physician needs to define all entities that they want to use as monitoring items (lines 7–12). In addition, it is mandatory that the physician specify the frequency of each new unit of thought, included in an element of a chronic component (lines 13–16). All entities defined in new chronic components are used to monitor the patient's chronic condition in subsequent consultations (line 17). Finally, the units of thought of the elements referenced in *chronic* components are marked as in use by applying *setUnitsInUse* procedure (lines 19–20).

---

**Algorithm 4** applyChronicCaseType(P, $CT^M_{chron}$)

---

1: A chronic case type $CT^M_{chron}$ = <name, components, true, chronicComponents> is selected

2: **if** name $\notin$ P.chronicCaseTypes **then**

3:　　elements ← getAllElementsIncludedIn(components)　　▷ First time of case type for patient P

4:　　setUnitsInUse(elements,$CT^M_{chron}$)

5: **else**

6:　　**if** chronicComponents = $\emptyset$ **then**

7:　　　　Evolution component $CC_{Evolution}$ emerges　　▷ Chronic components defined by physician

8:　　　　Physician defines all conceptual elements included in $CC_{Evolution}$

9:　　　　$CC_{others}$ can be defined to better monitor the chronic condition

10:　　　　$CCs_{new}$ = $CC_{Evolution} \bigcup CC_{others}$

11:　　　　newMonitorElems ← getAllElementsIncludedIn($CCs_{new}$)

12:　　　　newChronUnits ← getAllUnitsIncludedIn(newMonitorElems)

13:　　　　**for** newChronUnit **in** newChronUnits **do**

14:　　　　　　Physician needs to specify the frequency of newChronUnit

15:　　　　　　newChronUnit.ctSchedule is updated

16:　　　　**end for**

17:　　　　chronicComponents ← $CCs_{new}$ the chronic case type is updated

18:　　**end if**

19:　　elements ← getAllElementsIncludedIn(chronicComponents)

20:　　setUnitsInUse(elements,$CT^M_{chron}$)

21: **end if**

---

### 5.4. Usual Attention Flow

During an attention flow, a physician can take advantage of an already registered case type. Algorithm 5 shows how a case type can be used to record a frequent medical consultation scenario.

In Algorithm 5, a procedure waits until the physician selects a case type and applies it to the current consultation (lines 1–2). After a case type is applied, the physician can also make modifications in order to describe the accurate information of the entire clinical meeting (line 3). Each unit of thought marked as unique to the patient being evaluated is stored as personal data, and is removed from the current case type (lines 4–6). The algorithm continues by applying $T^*$ transformation, which generates a new medical record for the patient's history (lines 7–8). All chronic conceptual elements used in the case type are associated with the patient. Furthermore, if the case type is chronic, it is associated as permanent patient data (lines 9–13). Each unit of thought marked as unique to the current consultation is removed before updating the $CKB^M$ of the physician (line 14). To update $CKB^M$, the physician needs to specify if the current case type refers to a new workday scenario, or it is only an improvement over the previously selected case type (lines 14–21). Two data sets are modified after the usual attention flow: physician $CKB^M$ and patient history, including an $MR^P$ increment.

---

**Algorithm 5** Usual attention flow of a patient $P$

---

1: $CT^M \leftarrow$ selectSimilarCT()

2: applyCaseType(P, $CT^M$) is called

3: Physician M define $CT'^M$ by modifying the selected $CT^M$

4: personalInfo $\leftarrow$ getUqptUnits($CT'^M$)

5: P.personalData.add(personalInfo)

6: $CT'^M \leftarrow$ removeUqptUnits($CT'^M$) units marked with *uqpt* are removed

7: $mr_t^P \leftarrow T^*(CT'^M)$

8: $MR^P \leftarrow MR^P \bigcup \{mr_t^P\}$

9: chronElemts $\leftarrow$ getAllActiveChonicElementsInludedIn($CT'^M$)

10: P.chronicElems.add(chronElemts)

11: **if** isChronic($CT'^M$) **then**

12:     P.chronicCaseTypes.add($CT'^M$.name)

13: **end if**

14: $CT'^M \leftarrow$ removeUqcnUnits($CT'^M$) units marked with *uqcn* are removed

15: **if** $CT'^M$ is saved as an improvement **then**

16:     $CT^M \leftarrow CT'^M$

17:     $CKB^M$ is updated with the new version of $CT^M$

18: **else**

19:     $CT_{new}^M \leftarrow CT'^M$ is saved as a new case type

20:     $CKB^M \leftarrow CKB^M \bigcup \{CT_{new}^M\}$ the base is incremented

21: **end if**

---

*5.5. New Case Type Flow*

Algorithm 6 details the flow followed by the physician when they need to address a new case type that is not included in their *CKB*.

Since there is no case type to be re-used, Algorithm 6 needs to create an empty case type in which the new workday scenario can be detailed (lines 1–2). To define a new case type $CT_{new}^M$, the physician can re-use any predefined unit of thought, and can also create units of thought specifying new clinical phrases. Furthermore, predefined conceptual elements can be re-used and new elements can be created (lines 3–4). Each element defined by the physician is referenced from one clinical section. Therefore, new conceptual components are created in order to group elements sharing the same section type (lines 5–11). It is mandatory that the physician assigns a name to the new clinical

case type. The case type can also be marked as chronic, and in that case, the physician needs to specify the chronic attribute of each new element, created while defining the new case type (lines 12–21). All units of thought marked as unique to the patient are stored as personal data, and are removed from $CT_{new}^M$ (lines 22–24). Then, $T^*$ transformation is applied to generate a new medical record in the patient's history (lines 25–26). All chronic conceptual elements of $CT_{new}^M$ are associated with the patient, and if the case type is chronic it is associated as permanent patient data (lines 27–31). Finally, all units of thought marked as unique to current consultation are removed from the case type, and the clinical data base of the physician is enriched by including the new case type.

---

**Algorithm 6** New case type flow for the attention of patient $P$

---

1: There is no $CT^M$ selected by physician

2: $CT_{new}^M \leftarrow <$"new-name", $\varnothing$, false, $\varnothing >$ is created automatically

3: Physician creates new units $UTs_{new}$ and new elements $CEs_{new}$

4: Physician defines sections, by using $UTs_{new}$ and $CEs_{new}$ or pre-defined

5: secTypes $\leftarrow$ *ALL-SECTION-TYPES*

6: newComponents $\leftarrow \varnothing$

7: **for** secType$_i$ **in** secTypes **do**

8:      activeElems$_i \leftarrow$ [elementName, activeDescriptor] pairs in section$_i$

9:      $CC_{new_i} \leftarrow <$ maxCCId() + 1, secType$_i$, activeElems$_i >$

10:      newComponents $\leftarrow$ newComponents $\bigcup \{CC_{new_i}\}$

11: **end for**

12: Physician assigns a unique name to attribute *name* of $CT_{new}^M$

13: Physician can mark $CT_{new}^M$ as chronic

14: **if** $CT_{new}^M$ is marked as chronic **then**

15:      **for** elem **in** $CEs_{new}$ **do**

16:          Physician needs to specify the value of *elem.chron*

17:      **end for**

18:      $CT_{new}^M \leftarrow <$name, newComponents, true, $\varnothing >$

19: **else**

20:      $CT_{new}^M \leftarrow <$name, newComponents, false, $\varnothing >$

21: **end if**

22: personalInfo $\leftarrow$ getUqptUnits($CT_{new}^M$)

23: P.personalData.add(personalInfo)

24: $CT_{new}^M \leftarrow$ removeUqptUnits($CT_{new}^M$)

25: $mr_t^P \leftarrow T^*(CT_{new}^M)$

26: $MR^P \leftarrow MR^P \bigcup \{mr_t^P\}$

27: chronElemts $\leftarrow$ getAllActiveChonicElementsInludedIn($CT_{new}^M$)

28: P.chronicElems.add(chronElemts)

29: **if** isChronic($CT_{new}^M$) **then**

30:      P.chronicCaseTypes.add($CT_{new}^M$.name)

31: **end if**

32: $CT_{new}^M \leftarrow$ removeUqcnUnits($CT_{new}^M$)

33: $CKB^M \leftarrow CKB^M \bigcup \{CT_{new}^M\}$ the base is incremented

*5.6. Temporal Case Type Flow*

By applying a temporal case type, the history of patient $P$ is updated, and the $MR^P$ set is incremented with a new patient medical record. However, there is no change in physician $CKB^M$. Algorithm 7 details the use of a temporal case type.

---

**Algorithm 7** Temporal case type flow for the attention of patient $P$

---

1: $CT^M \leftarrow$ selectSimilarCT()

2: applyCaseType(P, $CT^M$) is called

3: Physician M define $CT'^M$ by modifying the selected $CT^M$

4: Physician M marks $CT'^M$ as a temporal case type

5: personalInfo $\leftarrow$ getUqptUnits($CT'^M$)

6: P.personalData.add(personalInfo)

7: $CT'^M \leftarrow$ removeUqptUnits($CT'^M$)

8: $mr_t^P \leftarrow T^*(CT'^M)$

9: $MR^P \leftarrow MR^P \cup \{mr_t^P\}$

10: chronElemts $\leftarrow$ getAllActiveChonicElementsInludedIn($CT'^M$)

11: P.chronicElems.add(chronElemts)

12: $CT'^M$ is deleted

---

Algorithm 7 is triggered after the physician assigns a temporal mark over an applied and modified case type (lines 1–4). As any other case type, all units marked as unique to the patient are stored as personal data, and are removed from the temporal case type (lines 5–7). $T^*$ transformation is also applied to create a new medical record (lines 8–9). Each chronic conceptual element referenced in the temporal case is permanently associated with the patient being evaluated (lines 10–11). The temporal case type is finally removed, since it is marked to be not re-used (line 12).

*5.7. Multiple Case Types Flow*

A physician can use more than one case type as the basis during the same medical consultation. Several rules are used to combine all conceptual components of each case type involved. To combine conceptual components, their active conceptual elements are accurately merged. The merge process takes into account the active elements described in the usual components, and active elements described in chronic components. Algorithm 8 details the method used to merge different case types.

Algorithm 8 starts by identifying the conceptual components included in each case type (lines 1–4). An ad hoc *merge* function is used to combine all identified components (line 5). Function *merge* is also applied over chronic conceptual components (lines 6–8). The algorithm continues by creating a case type $CT_{merge}^M$, which includes all merged components (lines 9–14). Then, the case type is applied and can be modified by the physician (lines 15–17). Each unit of thought marked as unique to the patient is taken into account as usual, it is stored as personal data and removed from the case type (lines 18–20). Likewise, a new medical record is created by applying $T^*$ transformation (lines 21–22). Each chronic conceptual element referenced in $CT_{merge}^M$ is permanently associated with the patient being evaluated, as well as any original chronic case type (lines 23–30). Finally, the used case type is deleted after concluding the consultation (line 31).

---

**Algorithm 8** Multiple case types flow for the attention of patient *P*

---

1: $CT_1^M \leftarrow$ selectSimilarCT()

2: $CT_2^M \leftarrow$ selectSimilarCT()

3: $comps_1 \leftarrow$ getAllComponents($CT_1^M$)

4: $comps_2 \leftarrow$ getAllComponents($CT_2^M$)

5: $comps_{merge} \leftarrow$ **merge**($comps_1$, $comps_2$)

6: $chronComps_1 \leftarrow$ getAllChronicComponents($CT_1^M$)

7: $chronComps_2 \leftarrow$ getAllChronicComponents($CT_2^M$)

8: $chronComps_{merge} \leftarrow$ **merge**($chronComps_1$, $chronComps_1$)

9: $name_{merge} \leftarrow$ concat($CT_1$.name,$CT_2$.name)

10: **if** $chronComps_{merge} \neq \varnothing$ **then**

11:     $CT_{merge}^M = <name_{merge}, comps_{merge},$ true, $chronComps_{merge} >$

12: **else**

13:     $CT_{merge}^M = <name_{merge}, comps_{merge},$ false, $\varnothing >$

14: **end if**

15: $CT_{merge}^M$ is auto-selected

16: applyCaseType(P, $CT_{merge}^M$) is called

17: Physician M define $CT_{merge}'^M$ by modifying $CT_{merge}^M$.

18: personalInfo $\leftarrow$ getUqptUnits($CT_{merge}'^M$)

19: P.personalData.add(personalInfo)

20: $CT_{merge}'^M \leftarrow$ removeUqptUnits($CT_{merge}'^M$)

21: $mr_t^P \leftarrow$ T*($CT_{merge}'^M$)

22: $MR^P \leftarrow MR^P \bigcup \{mr_t^P\}$

23: chronElemts $\leftarrow$ getAllActiveChonicElementsInludedIn($CT_{merge}'^M$)

24: P.chronicElems.add(chronElemts)

25: **if** isChronic($CT_1^M$) **then**

26:     P.chronicCaseTypes.add($CT_1^M$.name)

27: **end if**

28: **if** isChronic($CT_2^M$) **then**

29:     P.chronicCaseTypes.add($CT_2^M$.name)

30: **end if**

31: $CT_{merge}'^M$ is deleted

---

## 6. Instance-Based Learning

A learning method is proposed in order to generate suggestions for physicians. The proposed method is based on an ad hoc similarity metric, designed to compare the similarity between clinical case types.

### 6.1. Instance-Based Learning Method

An instance-based learning method is designed with the aim to provide suggestions for physicians. The proposed method takes into account the clinical knowledge base of a physician, in order to present suggestions based on previously defined case types. A register editor where a physician can take advantage of the proposed instance-based learning method is also introduced.

### 6.1.1. Register Editor

The register editor is an interface in which a physician can register a consultation appointment. The register editor presents personal information of the patient being evaluated, and includes an area for writing all details of a medical consultation. The main features of the register editor are illustrated in Figure 1, including a list of case type suggestions.

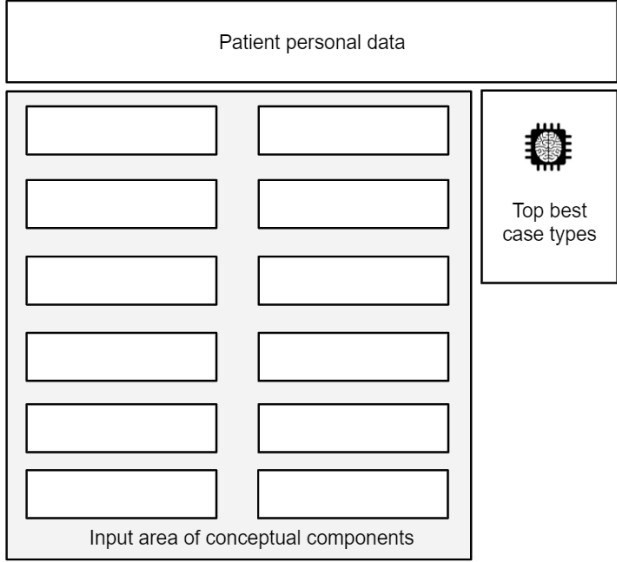

**Figure 1.** Main features of register editor.

The input area of the register editor is designed with the aim of registering a consultation in an organized structure, grouping information by clinical section types. When a physician writes in the input area, a case type $CT_{current}$ is automatically created, based on the information included in each section type. As a relevant feature, a list of similar case types is included in the register editor as suggestions for the physician. The suggested list is based on the top best values of a similarity metric, applied to compare the information of $CT_{current}$ against all case types previously registered.

### 6.1.2. Learning Method

A learning method is applied to determine the case types that best match with the clinical scenario of the patient being evaluated, according to an ad hoc similarity metric. A list of similar case types is suggested each time the physician modifies the information of the patient being evaluated. The list of similar case types is updated when introducing or removing any clinical phase during a medical consultation.

The proposed learning method implements a lazy approach [16], since the training stage of learning is delayed until a new case type draft must be evaluated. To evaluate a new case type draft, all previously defined case types are processed as training examples, and a similarity metric is applied to determine the most similar candidates. Algorithm 9 details how the instance-based learning method is implemented, seeking to suggest similar case types.

The learning method described in Algorithm 9 is triggered each time the physician modifies any aspect of the consultation being evaluated. An auxiliary case type $CT_{current}$ is created based on the information detailed by the physician in their register editor (line 1). Sentences without any meaningful word are not taken into account by the learning method (line 2). A step to remove duplicate units of thought is applied, since a physician could write duplicate clinical phrases in their register editor (line 3). Moreover, an array used to identify top best *similarity* metrics is initialized with empty values (line 4). The algorithm continues by iterating over all case types included in the physician clinical knowledge base (line 5). For each iteration, the similarity between $CT_{current}$ and any other case type

is calculated in order to update the array of top best metrics (lines 6–9). Finally, the case types with top best metrics are returned as suggestions to the physician (lines 11–12).

---

**Algorithm 9** Instance-based learning method

---

1:  $CT_{current} \leftarrow$ new CT(registerEditor.content)

2:  $CT_{current} \leftarrow$ removeNonMeaningfulUnits($CT_{current}$)

3:  $CT_{current} \leftarrow$ removeDuplicateUnits($CT_{current}$)

4:  topBest $\leftarrow$ Initialize array with $t$ empty values

5:  **for** $CT_i$ **in** $CKB^M$ **do**

6:    similarityMetric $\leftarrow$ *similarity*($CT_{current}$,$CT_i$)

7:    **if** similarityMetric is better that worstSimilarity(topBest) **then**

8:      topBest $\leftarrow$ replaceWorst(topBest,$CT_i$)

9:    **end if**

10: **end for**

11: topBest $\leftarrow$ removeEmptyValues(topBest)

12: **return topBest**

---

### 6.1.3. Using Suggested Case Types

A physician can select a suggested case type as a basis of writing a medical consultation. After a case type is applied, the input area of the register editor is updated by using the information defined in the selected case type. By using suggested case types, previously registered phases are re-used and the time spent writing the details of a consultation is reduced. Suggested case types can also remind physicians to verify important clinical aspects of their patients. Moreover, taking advantage of previously written sentences is useful when physicians need to address recurrent aspects of chronic patients.

### 6.2. Similarity Metric

A similarity metric is introduced in order to compare two case types of a clinical knowledge base. The proposed metric takes into account the similarity between conceptual components of different case types. Consequently, the similarity value between two case types is determined by the weighted similarities of their conceptual components.

### 6.2.1. Similarity Metric Definition

Sadegh-Zadeh [17] introduced the concept of *diagnostic relevance*, which applies fuzzy logic to evaluate the relevance of causal events associated with a clinical diagnosis. The proposed method is based on a similar idea, where the concept of *medical relevance* is considered to evaluate the relevance of conceptual components associated with a clinical case type.

Let $comp_{i,sec}$ be a conceptual component of case type $CT_i$ defined with $sec$ section type, where $sec$ belongs to ALL-SECTION-TYPES. The set ALL-SECTION-TYPES is introduced in Section 4 as a set that defines all possible clinical sections of the patient medical records. The similarity between two case types is denoted as *similarity*, and is defined by Equation (10).

$$similarity(CT_1, CT_2) = \sum_{sec} w_{sec} \times similarityCC_{sec}(comp_{1,sec}, comp_{2,sec}) \tag{10}$$

In Equation (10), each $w_{sec}$ defines the weight of a component with $sec$ section type. Consequently, the conceptual components of case types influence the *similarity* metric according to their $sec$ section type. The similarity weight of a clinical section type is determined by its medical relevance. The medical relevance is used to define the weight of each section type belonging to ALL-SECTION-TYPES = {$sec_1$, ..., $sec_n$}, as $W_{sec} = medRelevance(sec) / \sum_{sec_1}^{sec_n} medRelevance(sec_i)$.

The medical relevance of clinical section types must be defined based on background knowledge of the health area. Accurate weights of conceptual components provide a mechanism for reducing the impact of irrelevant features in the similarity metric [16]. As an example, health background knowledge suggests that the section for excuse notes should weigh less than the diagnosis section.

The following subsection presents the *similarityCC* function, introduced in Equation (10) for comparing two conceptual components of different case types.

### 6.2.2. Similarity between Components

To compare conceptual components, the similarity metric takes into account the units of thought included in all elements of conceptual components. The similarity between conceptual components is defined by Equation (11), which is aimed at comparing components sharing the same section type *sec*.

$$similarityCC_{sec}(cc_1, cc_2) = \begin{cases} 0, & \text{if } cc_1.secType \neq sec \vee cc_2.secType \neq sec \\ includedUTs(units(cc_1), units(cc_2)), & \text{otherwise} \end{cases} \quad (11)$$

Function *includedUTs*: $UT \times UT \rightarrow [-1,1]$ is applied to compare two sets of units of thought. Equation (12) presents *includedUTs* by considering different scenarios.

$$includedUTs(units_1, units_2) = \begin{cases} 0, & \text{if } units_1 = \varnothing \\ -1, & \text{if } units_1 \neq \varnothing \wedge units_2 = \varnothing \\ \max\{\sum_{u_1 \in units_1} \frac{belongs(u_1, units_2)}{|units_2|}, -1\}, & \text{otherwise} \end{cases} \quad (12)$$

If the first parameter $units_1$ of function *includedUTs* is an empty set, there is no unit of thought that can contribute as similarity data, then zero value is returned. Another exceptional case occurs when $units_2$ does not describe any information. If the second parameter is an empty set, the worst value of similarity must be returned because $units_1$ details clinical data not considered by $units_2$. A complex scenario arises when function *includedUTs* evaluates non-empty parameters. In that case, each unit of $units_1$ is analyzed in order to evaluate its inclusion into $units_2$, and a positive weight is determined for units that belong to both sets. In addition, a limit of maximum deference could be applied if $units_1$ and $units_2$ are significantly different and $units_1$ is bigger than $units_2$.

An auxiliary function *belongs(unit, units)* presented by Equation (13) is required to determine if a specific unit of thought belongs to a set of units. A *unit* that contradict the ideas represented by the *units* set is negatively weighted.

$$belongs(unit, units) = \begin{cases} 1, & \text{if } \exists\, u_{same} \in units \diagup equal(unit, u_{same}) \\ -1, & \text{otherwise.} \end{cases} \quad (13)$$

### 6.2.3. Similarity Metric Algorithm

The metric detailed in Algorithm 10 calculates the similarity of a case type $CT_{current}$ regarding any other case type. To achieve the final value of the similarity metric, Algorithm 10 needs to calculate similarity values of several conceptual components.

Algorithm 10 starts by initializing the similarity metric with a neutral value (line 2). Then, all section types of conceptual components included in analyzed case types are identified (line 3). After identifying the section types that influence the similarity metric, a relative weight factor is determined in order to accurately weigh the influence of each identified section type (line 4). Each section type with a positive weight of similarity is taken into account to calculate the value of the metric (lines 5–6).

---

**Algorithm 10** Similarity metric

---

1: function ***similarity***($CT_{current}$, $CT_i$)

2: similarity $\leftarrow$ 0

3: involvedSecTypes $\leftarrow$ sectionTypesOf($CT_{current}$) $\bigcup$ sectionTypesOf($CT_i$)

4: relativeWeight $\leftarrow$ relativeWeightFactor(involvedSecTypes)

5: **for** sec **in** involvedSecTypes **do**

6:     **if** sec.weight $\neq$ 0 **then**

7:         cachedSimilarityCC $\leftarrow$ getSimilarityCCValueFromCache(sec, $CT_i$)

8:         **if** cachedSimilarityCC is hitted **then**

9:             similarityCC $\leftarrow$ cachedSimilarityCC

10:         **else**

11:             $units_{current}$ $\leftarrow$ getUnitsBySectionType(sec,$CT_{current}$)

12:             $units_i$ $\leftarrow$ getUnitsBySectionType(sec,$CT_i$)

13:             **if** $units_{current}$ $\neq$ $\varnothing$ **then**

14:                 **if** $units_i$ $\neq$ $\varnothing$ **then**

15:                     includedUTs $\leftarrow$ 0

16:                     **for** $unit_{current}$ **in** $units_{current}$ **do**

17:                         belongs $\leftarrow$ *belongs*($unit_{current}$, $units_i$)

18:                         **if** belongs **then**

19:                             includedUTs $\leftarrow$ includedUTs $+ \frac{1}{|units_i|}$

20:                         **else**

21:                             includedUTs $\leftarrow$ includedUTs $- \frac{1}{|units_i|}$

22:                         **end if**

23:                     **end for**

24:                     similarityCC $\leftarrow$ max{includedUTs, $-1$}

25:                 **else**

26:                     similarityCC $\leftarrow$ $-1$

27:                 **end if**

28:             **else**

29:                 similarityCC $\leftarrow$ 0

30:             **end if**

31:             putSimilarityCCValueInCache(similarityCC,sec,$CT_i$)

32:         **end if**

33:         similarity $\leftarrow$ similarity + (relativeWeight * similarityCC)

34:         bestRemain $\leftarrow$ upperBound(sec, involvedSecTypes)

35:         **if** similarity + bestRemain $<$ worstSimilarity(*topBest*) **then**

36:             *invalidateSimilarityCCValuesOnCache($CT_i$)*

37:             **throw** *discard-low-similarity*

38:         **end if**

39:     **end if**

40: **end for**

41: **return** similarity

---

For each identified section type, the similarity of components sharing the same section type must be calculated. A cache containing values of similarity between conceptual components is used to improve the performance of the proposed metric (lines 7–9). To calculate the similarity between conceptual components, the sets of units of thought included in each component are determined. Algorithm 10 implements the rules introduced by Equation (12) for calculating the similarity between two sets of units of thought, including the general scenario (lines 15–24) for non-empty sets, and exceptional scenarios (line 26 and line 29) to address singular situations of empty sets. Moreover, the calculated value of component similarity is cached, to be used in the future (line 31). The partial value of the similarity metric is updated after calculating the similarity between each pair of conceptual components. For each pair of components, the partial similarity is affected by the similarity of the components according to a relative weight factor (line 33).

To detect low values of similarity, an upper bound is calculated in order to determine a maximum possible value of similarity (line 34). If the similarity between two case types is detected early as low, it is not required to calculate its exact value. All case types with low *similarity* are discarded early, and their partial values of component similarity are removed from the cache as they are not fully calculated (lines 35–37). At last, a final value of similarity is returned after iterating over all involved section types (line 41).

### 6.3. Implementation of Similarity Metric

The similarity metric is an essential feature of the proposed learning method. The metric must be able to accurately compare the similarity between clinical case types, and it also needs to execute as quickly as possible. The similarity metric is highly demanded in virtue of the lazy approach of the learning method. Therefore, several techniques of indexing and cache are applied for reducing the metric execution time. All optimizations implemented to reduce the execution time of the *similarity* metric are presented in the following paragraphs.

*Compare units by canonical form.* The operator *equal* for units of thought is used to determine if two different sentences represent the same clinical idea. To implement the *equal* operator, a canonical transformation is applied over the units being compared. Two transformations are applied by comparing a pair of units of thought. For each unit of thought, structured information and random data are removed, in order to achieve the canonical form. Finally, a raw string comparison between both canonical forms is evaluated. Original units of thought are identified as equal only if they coincide in their canonical form.

*Zero similarity value.* Function *similarity($CT_1$, $CT_2$)* is called to calculate the similarity between a case type $CT_1$ and another case type $CT_2$. Both case types are composed by conceptual components that influence the similarity metric according to its section type weights. However, an empty component of $CT_1$ cannot provide similarity information since it does not have associations with units of thought. If a conceptual component of $CT_1$ is empty, its similarity in regard to any other component is zero. No calculation is performed over the empty components of $CT_1$, rather all computational effort is performed over its non-empty components.

*Comparing with empty components.* All components of a case type $CT_1$ are analyzed when calculating the similarity of $CT_1$ in regard to another case type $CT_2$. Each conceptual component of $CT_1$ should be compared against a component of $CT_2$ with the same section type. If $CT_2$ does not include a conceptual component with the same section type, then a value representing the biggest difference of similarity is returned without performing additional calculations.

*Cache of previous similarity values.* The proposed similarity metric provides a mechanism for comparing different case types. However, the metric is not based on case types themselves, but on their conceptual components. Due to the high need of obtaining similarities between conceptual components, a cache is designed for containing pre-calculated values of component similarities. Figure 2 shows the structure used to maintain recent values of similarities, and how similarity values are cached for each case type included in

the clinical knowledge base of a physician. The proposed structure is able to cache the last value of similarity of all conceptual components of each case type.

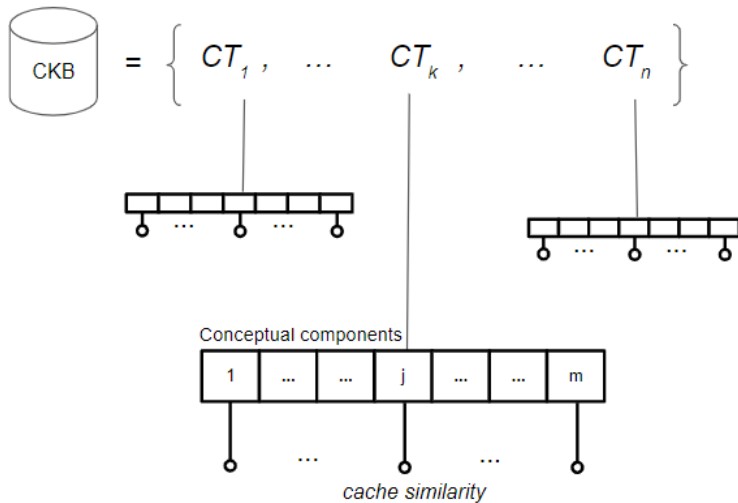

**Figure 2.** Cache structure for similarity between components. The clinical knowledge base is composed of case types, each one containing similarity cached values of its conceptual components.

After evaluating the similarity between a specific case type in regard to any other case type, all values of component similarities are stored in the cache. Figure 3 introduces a scenario in which a "Case type A" is slightly modified, by only changing the information described in one of its conceptual components. Several highlighted values of component similarities are obtained from the cache. Furthermore, a high cache hit ratio should be achieved after re-using any other case type and applying a few modifications.

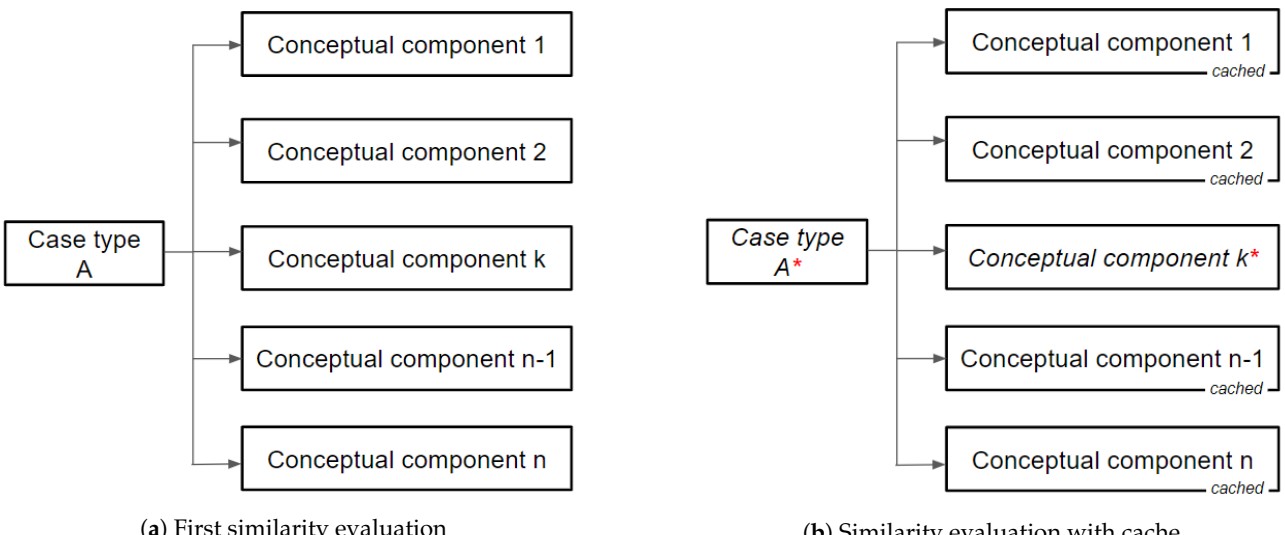

(**a**) First similarity evaluation          (**b**) Similarity evaluation with cache

**Figure 3.** Use of similarity cache values.

*Discard non-promising candidates.* The proposed learning method is designed to suggest the best case types that can be applied during a medical consultation. Top best case types are identified according to best *similarity* metric values, and only *t* best case types are presented to the physician.

The similarity function separates the first *k* section types from the rest of the ALL-SECTION-TYPES set, as described in Equation (14).

$$similarity(CT_1, CT_2) = \sum_{sec=sec_1}^{sec_N} w_{sec} \times similarityCC_{sec}$$

$$= \sum_{sec=sec_1}^{sec_k} w_{sec} \times similarityCC_{sec} + \sum_{sec=sec_k+1}^{sec_N} w_{sec} \times similarityCC_{sec} \qquad (14)$$

$$\leq \sum_{sec=sec_1}^{sec_k} w_{sec} \times similarityCC_{sec} + \sum_{sec=sec_k+1}^{sec_N} w_{sec} = \sum_{sec=sec_1}^{sec_k} w_{sec} \times similarityCC_{sec} + R_{constant}^{k+1}$$

Equation (15) presents an upper bound inferred by simplifying (14), which can be used for discarding case types with low similarity values.

$$similarity(CT_1, CT_2) \leq \sum_{sec=sec_1}^{sec_k} w_{sec} \times similarityCC_{sec} + R_{constant}^{k+1} \qquad (15)$$

Several component similarity values (*similarityCCs*) need to be computed to obtain the final value of *similarity(CT$_1$, CT$_2$)*. An upper bound is identified after determining the value of *similarityCC$_{sec_k}$*. After calculating the similarity of the *k*th conceptual component, it is possible to use an upper bound to discard a case type with a low similarity value. Each case type whose upper bound of similarity is lower than the worst element of the top best metrics is considered a non-promising candidate, and no more computational effort is expended to calculate its final similarity value.

## 7. Experimental Validation

This section presents the experimental validation of the proposed approach on a real case study, which served as a basis for evaluating the practical aspect of this research.

### 7.1. Problem Instances

The source *Clinical cases in primary care* [18] was used for evaluating the proposed approach. The source is a multi-authored publication that covers a wide range of clinical scenarios of primary care.

#### 7.1.1. Prerequisites for Building Case Type Instances

The collaboration of advanced medical students was requested with the intention of registering as many clinical scenarios as possible. Students were instructed to record the primary care scenarios described in *Clinical cases in primary care* as new clinical CTs. In order to group all the information recorded, it was necessary to implement procedures for exchanging clinical CTs. The *export* and *import* procedures were used to exchange different CTs.

A procedure to *export* a given CT was implemented. The procedure extracts a CT from a specific clinical knowledge base CKB, and it also anonymizes any information that refers to the person who wrote (owner) the CT. The *import* procedure consolidates the information of a specific CT into a target CKB. A new CT is inserted into the target CKB, replacing any anonymized reference of the original owner with the person who owns the target CKB. Importing a CT is a complex procedure, which must avoid the generation of duplicate units of thought, and has to merge the conceptual elements of the new CT with those existing in the target CKB.

#### 7.1.2. Building Case Type Instances

The set of clinical cases specified in the publication *Clinical cases in primary care* was distributed to be evaluated by 50 advanced medical students. Each student had to evaluate three different cases, and each clinical case was assigned to at least one student. Furthermore, each student was instructed to contribute two additional clinical cases, defined as variants of those presented in the clinical source.

All scenarios of primary care detailed in the clinical source were successfully registered by the group of students, including variants of repeated clinical scenarios. Algorithm 11 details how a single CKB was loaded with 250 scenarios of primary care, based on information registered by students.

---

**Algorithm 11** Building case types

---

1: **for** i = 1 **to** length(STUDENT-LIST) **do**

2:     $student_i \leftarrow$ STUDENT-LIST[i]

3:     $CKB_{student_i} \leftarrow \emptyset$

4:     **for** j = 1 **to** 3 **do**

5:         *k-index* $\leftarrow$ mod(3(i − 1) + j, length(CASE-SOURCE))

6:         $CT_{i_j} \leftarrow student_i$ records case number *k-index* of CASE-SOURCE

7:         $CKB_{student_i} \leftarrow CKB_{student_i} \bigcup \{CT_{i_j}\}$

8:     **end for**

9:     $CT_{i_{v_1}} \leftarrow$ first variant of case type included in $CKB_{student_i}$

10:     $CKB_{student_i} \leftarrow CKB_{student_i} \bigcup \{CT_{i_{v_1}}\}$

11:     $CT_{i_{v_2}} \leftarrow$ second variant of case type included in $CKB_{student_i}$

12:     $CKB_{student_i} \leftarrow CKB_{student_i} \bigcup \{CT_{i_{v_2}}\}$

13: **end for**

14: $CKB_{all} \leftarrow \emptyset$

15: **for** i = 1 **to** length(STUDENT-LIST) **do**

16:     $student_i \leftarrow$ STUDENT-LIST[i]

17:     **for** j = 1 **to** 5 **do**

18:         $CT_{i_{jexported}} \leftarrow export(j, CKB_{student_i})$

19:         $import(CT_{i_{jexported}}, CKB_{all})$

20:     **end for**

21: **end for**

22: **return** $CKB_{all}$

---

Algorithm 11 starts by initializing all CKBs of the students (STUDENT-LIST) selected for recording new CTs (lines 1–3). Each student is expected to treat three fictitious patients suffering from one of the specific conditions of the clinical source (lines 5–6). Moreover, two variants contributed by each student are also registered (line 9 and line 11). Therefore, the CKB of each student is enriched with five new CTs (line 7, line 10, and line 12). The algorithm continues by initializing a single $CKB_{all}$ that groups all information recorded by all students (line 14). Each registered CT is exported using the *export* procedure, and the *import* procedure is applied to consolidate the exported CT into the $CKB_{all}$ (lines 15–21). Finally, the $CKB_{all}$ which contains all the 250 registered CTs (five contributed by each of the 50 students) is returned (line 22).

### 7.2. Parameter Settings of Similarity Weight

For the purposes of the experimental evaluation, the set of clinical section types was defined following the Uruguayan health model. The set of clinical section types was defined as *ALL-SECTION-TYPES* = {*Diagnosis, Consultation reason, Current illness, Physical examination, Medication, Studies, Procedures, Referral, Message agents, Advisors, Excuse notes, Observations*}.

The similarity weight of a clinical section type is given by its medical relevance. A simple medical relevance criteria was applied to give greater weight to the most important section types. Four levels of importance were defined in order to consider qualitative

ranges of medical relevance. The level scale used to define medical relevance was: *very important*, *fairly important*, *important*, and *slightly important*. Table 2 presents the weight values of the clinical section types used in the experimental evaluation, grouped by level of medical relevance. Table 2 shows that the weight of the diagnosis section was defined with a high value of $W_{Diagnosis} = 0.16$. On contrary, the excuse notes section was defined with a lower weight of $W_{Excuses} = 0.04$.

**Table 2.** Weight of conceptual component types.

| Very Important (Weight 0.16) | Fairly Important (Weight 0.12) | Important (Weight 0.08) | Slightly Important (Weight 0.04) |
|---|---|---|---|
| Diagnosis | Consultation reason<br>Current illness<br>Physical examination | Medication<br>Studies<br>Procedures<br>Referral | Message agents<br>Advisors<br>Excuse notes<br>Observations |

All weights presented in Table 2 influence the calculation of the similarity metric of the learning method. Equation (16) shows the consistency of presented weights used for the similarity metric.

$$\sum_{sec_1}^{sec_n} W_{sec} = \sum_{very\ important} W_{sec_v} + \sum_{fairly\ important} W_{sec_f} + \sum_{Important} W_{sec_i} + \sum_{slightly\ important} W_{sec_s}$$
$$\sum_{sec_1}^{sec_n} W_{sec} = 0.16 + 3 \cdot 0.12 + 4 \cdot 0.08 + 4 \cdot 0.04 = 1 \quad (16)$$

### 7.3. Performance Evaluation

An experimental evaluation was conducted in order to analyze the lazy nature of the proposed learning method. In the learning approach, a similarity metric between clinical CTs is calculated by using all previously recorded CTs as training examples. Since the problem-solving ability of the proposed method is increased with each newly defined CT, it is important to analyze the performance of the proposed learning method when faced with CKBs with a great number of CTs.

#### 7.3.1. Execution Platform of Performance Evaluation

The execution time analysis was performed on an Intel(R) Core(TM) i7-4700MQ CPU @ 2.40 GHz, 16 GB RAM, and running 64-bit Windows 10 Pro.

#### 7.3.2. Execution Time

The efficiency of the learning method was evaluated when faced with CKBs of different sizes. To make a realistic evaluation, the 250 CTs registered by students were considered as input data, and the average time of 50 executions was measured for each CKB analyzed.

Figure 4 presents the average execution time of the proposed method when using different CKB sizes. The algorithm was executed on CKBs containing from 25 to 250 CTs.

Figure 4 shows how the size of a CKB has a direct influence on the execution time of the proposed method. Results also demonstrate that the proposed learning method is able to process 250 CTs in less than 90 milliseconds.

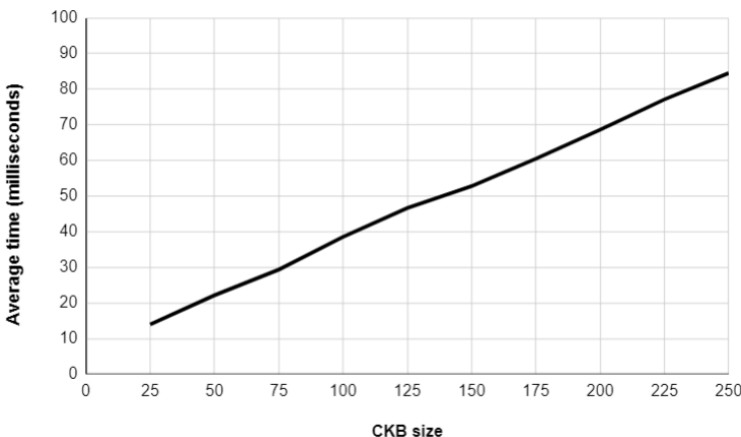

**Figure 4.** Average execution time of the proposed learning method regarding different CKB sizes.

### 7.3.3. Execution Time Projection

In order to estimate the efficiency of the learning method when facing larger CKBs, auxiliary CTs were generated based on the information recorded by the students. Although the auxiliary CTs were artificially built and do not reflect real clinical scenarios, they can be used to evaluate the performance of the learning method as they have the same data dimension as the CTs written by the students. The graphic in Figure 5 reports the average time of the proposed method regarding CKB sizes.

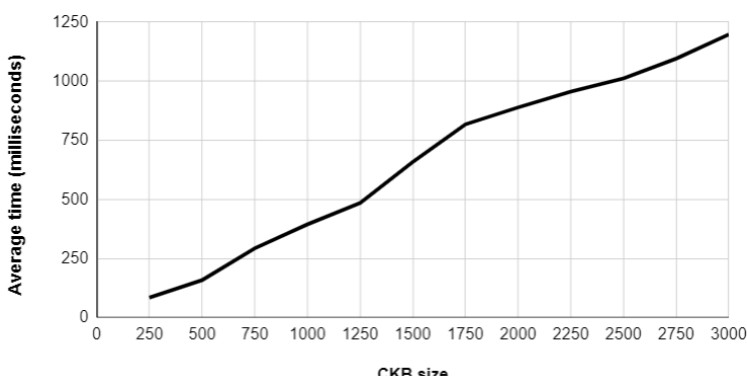

**Figure 5.** Average execution time of the proposed learning method when facing larger CKBs.

Figure 5 shows that the learning method generated suggestions in less than 1.25 s, even when facing larger CKBs with a great number of CTs. Given that 3000 represents a suitable bound for the number of CTs included in a physician CKB, the proposed method is able to produce suggestions in reasonable execution times, even when processing real CKBs with several workday scenarios.

### 7.3.4. Comparison with a Bayesian Learning Approach

To further analyze the applicability of the proposed approach, this subsection presents a comparison of the proposed instance-based learning method with a Bayesian learning method, which is based on a classical algorithm described by Mitchell [16] for classifying text documents.

The implemented Bayesian learning method works under the assumption that the occurrence probability of a word is independent of its position within a document. During the learning task, all medical records are examined as training examples, aiming at extracting the vocabulary of all words appearing in patient histories. After that, the frequency of each word is computed on all case types, to obtain the probability estimates of the Bayesian approach. Finally, to classify a new draft of the register editor, the probability estimates are used to determine the most likely case type to be applied.

Figure 6 reports the average execution time of both the Bayesian method and the instance-based method, regarding different CKB sizes, when processing the testbed of 250 CTs registered by students.

The graphic in Figure 6 uses a logarithmic scale to highlight the difference of two orders of magnitude between the execution times of both learning methods. The efficiency results reveal that the Bayesian learning method is difficult to apply due to high execution times, even when processing small volumes of data. Execution time results also reaffirm the benefits of the instance-based learning method, which significantly outperforms the Bayesian approach in terms of efficiency.

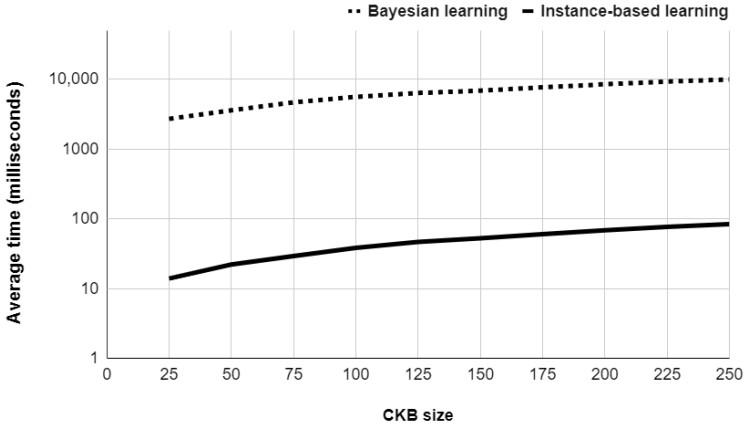

**Figure 6.** Average execution time comparison: instance-based learning vs. Bayesian learning method.

### 7.4. Testing the Applicability of the Instance-Based Learning Approach

In order to test the applicability of the proposed approach, a prototype was developed and deployed on Google Compute Engine, the Infrastructure as a Service component of the Google Cloud Platform. The prototype was evaluated by advanced medical students in their last year of training at Universidad de la República, Uruguay.

### 7.4.1. Comparison with Praxis

Praxis reports the average time required to write a CT starting from an empty CKB [15]. In order to compare the proposed approach with the original implementation of Praxis, the average time to write a CT using the prototype was measured. Figure 7 illustrates both average writing times starting from an empty CKB, by considering the medical attendance of the first 50 patients.

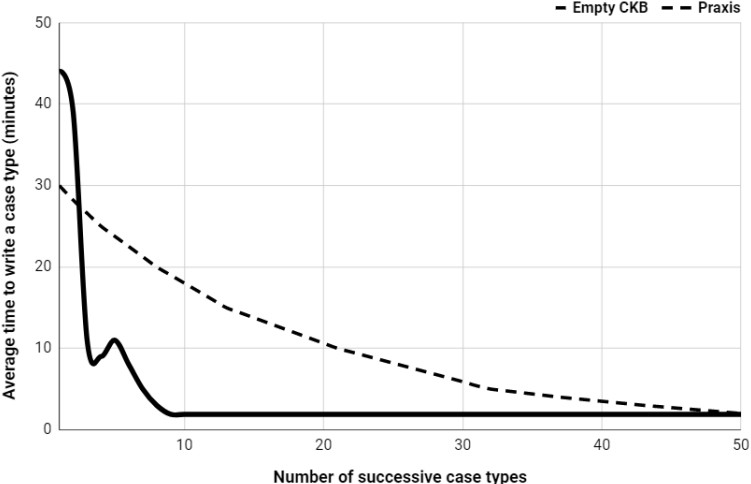

**Figure 7.** Average time of 50 medical students to write the notes of a case type (continuous line). Average time according to Praxis reports (dotted line). Both evaluations start with an empty CKB.

Although Praxis presents shorter registering times for the first two medical consultations, more than 50 evaluations are needed to achieve a convergence point. The proposed approach significantly reduces registration times from the third case registered onwards, converging quickly to less than three minutes of writing consultations. The proposed learning method demanded 210 min to register 50 consultations (i.e., 4.2 min per consultation), while using Praxis requires 519 min (10.4 min per consultation). The overall time reduction factor is 2.5×.

### 7.4.2. Improvement Using a Pre-Loaded CBK

The time needed to register a medical consultation can be reduced by using previously registered information. The average time to record a CT was measured in a context in which medical students could use a pre-loaded CKB. Figure 8 shows the average time to write a CT taking advantage of a pre-loaded CKB containing typical workday scenarios. As a relevant result, the use of a pre-loaded CKB implied a reduction of up to five minutes for recording the notes of the first six medical consultations. Furthermore, a pre-loaded CKB also accelerated the convergence to three minutes of writing consultations.

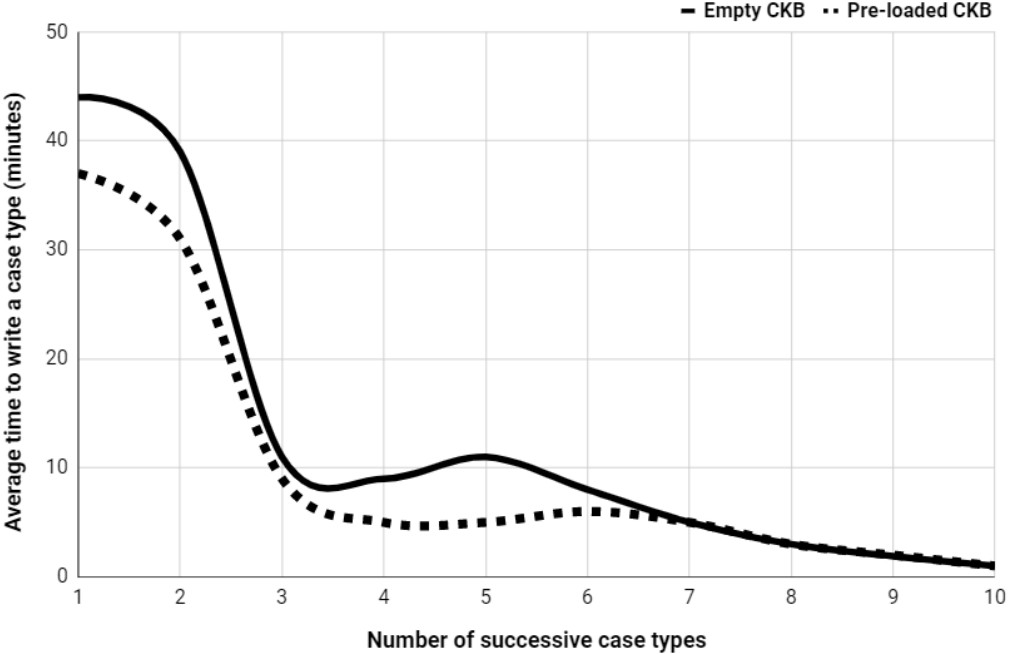

**Figure 8.** Average time of 50 medical students starting with an empty CKB (continuous line). Average time of 50 medical students taking advantage of a pre-loaded CKB (dotted line).

Regarding the scalability of the incremental processing of new case types, results suggest a convergence towards a short writing time for medical consultations, even when processing large volumes of data.

### 7.4.3. Survey about the Proposed Approach

More than 50 medical students from different editions of the Medical Informatics course were surveyed after using the prototype of the proposed approach. The advanced medical students have tested the prototype during course editions from 2016 to 2020. Figure 9 summarizes the best features identified by students.

Results show that 43% of the surveyed students mentioned that the learning curve was steep before they could benefit from the proposed learning method. As a relevant result, more than 73% of the students considered the prototype as an appropriate tool for medical practice, especially at medical consultations. Moreover, 62% of the students were able to speed up writing time during medical consultations.

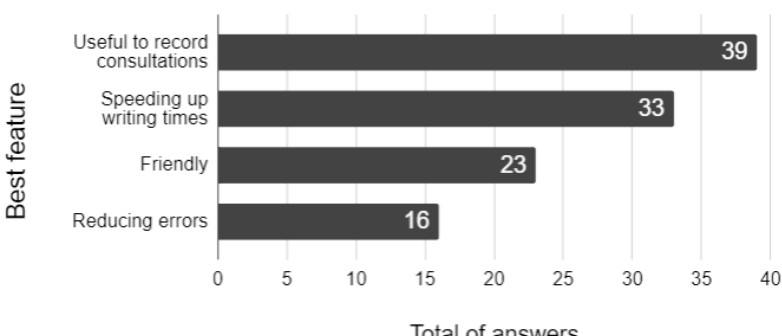

**Figure 9.** Best features of the proposed approach, according to the survey performed on students.

### 7.5. Interoperability of Health Information

Health terminological standards were taken into account due to the relevance of the interoperability of information in the Medical Informatics area. In particular, the national drug dictionary of Uruguay and a terminology server provided by the Hospital Italiano de Buenos Aires (HIBA) were integrated into the proposed approach.

### 7.5.1. Integrating the National Drug Dictionary of Uruguay

A National Drug Dictionary (DNMA) is defined by *Salud.uy* in order to standardize the information and vocabulary for clinical and logistical use applied to pharmaceutical and related products. DNMA acts as a standard of drug reference terminology for the network of healthcare service providers in Uruguay. Access permissions were requested from the DNMA in order to import the national dictionary of medicines into the proposed approach. Importing the national drug dictionary helped build a functional model, in which physicians can make pharmacological indications using a wide range of drugs.

### 7.5.2. Using the Terminology Server of Hospital Italiano de Buenos Aires

A terminology server allows linking the free text entered by a physician in a medical record to different health classifications, such as ICD9-CM, ICD10, or LOINC [4]. The use of a terminology server allows clinical information to be recorded in a structured form, using clinical terminology standards. Terminology standards enable interoperability of clinical information, and also allow information to be re-used for other purposes, such as clinical decision support, data analysis, and research.

The proposed system is able to use the terminology server supported by HIBA. The terminology server publishes its terminological terms grouped in different domains. This work has been successful in using terminology services for the domains that cover: reasons for consultation, diagnoses, procedures, and studies, which are required for the Uruguayan medical records model.

## 8. Discussion

The experimental evaluation of the proposed instance-based learning method focused on the practical aspect of the research. Thus, the evaluation was performed on a real use scenario, where the proposed approach demonstrated advantages over the original implementation of Praxis. Additionally, results were better in terms of writing times when using a pre-loaded CKB, containing typical workday clinical scenarios. Regarding the usability of the proposed system, a survey performed on a group of advanced medical students showed a high rate of approval. The implemented prototype was highlighted as an appropriate tool for medical practice and useful at medical consultations. Furthermore, and despite the lazy nature of the proposed method, the results showed that the learning approach was able to produce suggestions in reasonable execution times, even when dealing with large volumes of data.

Specific strategies can be applied to reduce uncertainties, including using expert knowledge to design and generate useful realistic instances for learning, and expanding

the similarity metrics considered for the comparison of clinical information. In this regard, a recommendation for the practical aspect of the research is gathering and organizing as much information as possible about clinical practice in a systematic way, in order to help the automatic system to expand its base of knowledge to generate more accurate suggestions. In turn, a physician should be properly trained to register all the relevant data for the proposed learning-based system, without omitting any important information.

In this context, the main research lines for future work are related to evaluate the proposed system in a professional work environment of healthcare attention, with the aim of improving the accuracy of the learning method based on professional feedback. Thus, a future work line includes studying the proposed approach with the help of professional physicians. Another possibility for future work is related to enhance the accuracy of the learning method by improving the comparison between units of thought (clinical phrases). The weights of the clinical sections of case types used in the experimental analysis were defined simply, according to qualitative ranges of medical relevance. Consequently, a future work is to enhance the results by considering more accurate weights of medical relevance.

## 9. Conclusions

This work presented a novel approach to represent clinical knowledge, which supports an appropriate methodology for recording medical consultations. An instance-based learning method was also proposed, aiming at providing pertinent suggestions for physicians. Different scenarios of medical consultations were modeled to address the diversity of situations of physician workday, including multiple diagnoses and the attention of chronic patient. The proposed formal structure was also designed to use standard health terminology and codification. The approach was validated on a real case study involving 250 real instances constructed by advanced medical students. The proposed instance-based learning method was able to generate suggestions in reasonable execution times, even faced with large volumes of data. A total of 62% of the participants reduced the writing time of their medical consultations, which demonstrated that the approach was useful to accelerate the clinical registration process. Furthermore, results indicated it was appropriate to follow physician reasoning, especially during medical consultations. More than 73% of the participants approved a prototype following the proposed approach for assistance during consultations.

The proposed clinical representation supported by the learning method contributed to generate medical records faster than when using mainstream EMR systems. Overall, the proposed approach is a first step to explore new ways to foster physician thinking, overcoming difficulties of template-based clinical systems that are not designed from the medical point of view.

**Author Contributions:** Conceptualization, M.G. and F.S.; methodology, M.G.; software, M.G.; validation, M.G. and F.S.; formal analysis, M.G. and S.N.; investigation, M.G. and S.N.; resources, M.G. and F.S.; data curation, M.G.; writing—original draft preparation, M.G. and S.N.; writing—review and editing, S.N.; visualization, S.N.; supervision, F.S. and S.N.; project administration, M.G.; funding acquisition, M.G., F.S. and S.N. All authors have read and agreed to the published version of the manuscript.

**Funding:** This research was partially funded by ANNI, Uruguay and Infor-Med Corporation, Argentina. Part of the publication fee was funded by PEDECIBA, Uruguay. The work of S. Nesmachnow was partly funded by ANII and PEDECIBA, Uruguay.

**Conflicts of Interest:** The authors declare no conflict of interest.

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
