# Peer review of "Instance-Based Learning Following Physician Reasoning for Assistance during Medical Consultation"

_applsci, doi:10.3390/app11135886_

Round 1

Reviewer 1 Report

Please, find the attached file.

Author Response

Description of changes performed to meet reviewers’ requests to the manuscript "Instance-based learning following physician reasoning for assistance during medical consultation", Applied Sciences, 2021

Dear Editor-in-Chief and reviewers,

Thank you for handling/reviewing our submission. We have modified our manuscript according to the comments made by the reviewers and we have prepared a detailed response to each issue pointed out by each reviewer. We have submitted the updated manuscript, where new content is marked in blue font. The answers to reviewers are provided in this letter.

We improved our article taking into account all the comments made by the reviewers. The main changes in the manuscript included:

  • The abstract was re-written to better describe the content of the article and the quantitative results.
  • We included a comparison with a Bayesian learning approach, such as the one applied in previous works.
  • We have included a new section to present specific comments about the usability of the proposed method.
  • We have included proper references and further explained some important concepts about the proposal.

In our opinion, by following the suggestions made by the reviewers, the structure, readability, and scientific contribution of the manuscript has been greatly improved.

The reviewers comments are copied below and our answers are detailed next. New content has been marked in blue font in the revised version of the article.

Reviewer #1 

Comment:
# Overall statement or summary of the article: This article presents an automatic system for modeling clinical knowledge to follow physicians reasoning in medical consultation. The topic for this study is interesting and sound, and the study may be useful for clinical applications; However, there are several major corrections that need to be considered before further processing.
Response: Thanks for your comments. 

Comment:
# The authors need to rewrite the abstract and add some of the most important quantitative results.
Response: Thanks for the observation. Following your suggestion, we have re-written the abstract and included a better description of the main contributions, the new comparison with a Bayesian learning approach and the main quantitative results. Further explanation about the quantitative results have been included in Section 6.

Comment: 
# In section 3, the authors should clearly mention the weakness point of former works (identification of the gaps) and shows the key differences between the different previous methods and the proposed method. A comparative overview table should solve this point.
Response: Thanks for your comments, we agree that including the comparison and the suggested table improves the description of related works. In that section, we included a new table (current Table 1), showing the main details of related works (relevant features and identified weaknesses). In turn, we have included new content in page 4 to describe the key differences of existing approaches and our proposal.

Comment:
# please provide reference/references for this sentence (Unfortunately, conventional EMR systems are template-based products that generate poor quality data, due to long search mechanisms and excessive mandatory fields, which often add noise to the relevant patient information) in section 2. 
Response: We have included a proper reference that supports the the assertion that EMR systems are not flexible, therefore they usually generate poor quality data (i.e., article “Terminology Services: Standard Terminologies to Control Health Vocabulary” by González et al., published in Yearbook of Medical Informatics (2018).

Comment:
# Please provide a reference for Equation 3 and 4. 
Response: Equations 3 and 4 are original from our article. Thus, there is no reference to include.

Comment:
# The proposed approach (lazy approach mentioned in section 6.1.2) isn't defined in detail (e.g., no mathematical equations and no justifications for design choices). Please provide the necessary information. 
Response: Thanks for the observation. Indeed, in the original version of the article the lazy nature of the proposed methods was not clearly described. In the revised version, we have properly described the main concepts of the lazy approach in the description of the proposed method in Section 6.1.2. In a nutshell, the proposed method is lazy because the training phase is delayed until a new case type draft must be evaluated.

Comment:
# In the section 7.3.2, the authors should add results comparison between the execution time of the proposed method and the execution time from other techniques that will be very worth in this section. 
Response: Thanks for the suggestion. For the revised of the article we have developed a method based on Bayesian learning, following the approach by Mitchell (1997). In the new section 7.3.4 we have described the Bayesian learning method used as a reference baseline, we have reported the comparison of execution times and finally some comments about the benefits of the proposed method, which significantly outperforms the Bayesian approach in terms of efficiency.

Comment:
# Please replace the summary of results (section 7.6) with a section named discussion (section 8) and provide a brief discussion about a usability of proposed method, strategies or recommendations to reduce uncertainties. The author should also move the future work from the conclusion (section 8.2) to the discussion section. The conclusion section prefer to be as one paragraph and renamed (section 9).
Response: Following your suggestion, we have included a new section (Section 8: Discussion) with specific comments about the usability of the proposed method. We also included details about strategies or recommendations to improve the results and reduce uncertainties. The future work was also moved from the conclusion section to new Section 8. The conclusion section was shortened to one paragraph, as recommended (an additional overall comment is included to end the article).

Reviewer 2 Report

Very nice paper, I greatly appreciated the proposition and the effort.

Just I noted a few typos:

  • line 181 "schedule" should read "ctSchedule" instead
  • line 232 "model" should read "modeled" instead
  • line 329 "no" should read "not

Furthermore:

I feel the author should try to cite Handbook of Analytic Philosophy of Medicine by Kazem Sadegh-Zadeh. I feel the value of the paper will be improved in the part regarding the logic of medical diagnosis and the fuzzy logic approach of Sadegh-Zadeh which seems, by the way, reinvented in the similarity metric definition (paragraph 6.2.1 from line 424).

Author Response

Description of changes performed to meet reviewers’ requests to the manuscript "Instance-based learning following physician reasoning for assistance during medical consultation", Applied Sciences, 2021

Dear Editor-in-Chief and reviewers,

Thank you for handling/reviewing our submission. We have modified our manuscript according to the comments made by the reviewers and we have prepared a detailed response to each issue pointed out by each reviewer. We have submitted the updated manuscript, where new content is marked in blue font. The answers to reviewers are provided in this letter.

We improved our article taking into account all the comments made by the reviewers. The main changes in the manuscript included:

  • The abstract was re-written to better describe the content of the article and the quantitative results.
  • We included a comparison with a Bayesian learning approach, such as the one applied in previous works.
  • We have included a new section to present specific comments about the usability of the proposed method.
  • We have included proper references and further explained some important concepts about the proposal.

In our opinion, by following the suggestions made by the reviewers, the structure, readability, and scientific contribution of the manuscript has been greatly improved.

The reviewers comments are copied below and our answers are detailed next. New content has been marked in blue font in the revised version of the article.

Reviewer #2

Comments:

Very nice paper, I greatly appreciated the proposition and the effort.

Just I noted a few typos:

  • line 181 "schedule" should read "ctSchedule" instead
  • line 232 "model" should read "modeled" instead
  • line 329 "no" should read "not

Response: All typos have been corrected in the revised version of the article.

Furthermore:
I feel the author should try to cite Handbook of Analytic Philosophy of Medicine by Kazem Sadegh-Zadeh. I feel the value of the paper will be improved in the part regarding the logic of medical diagnosis and the fuzzy logic approach of Sadegh-Zadeh which seems, by the way, reinvented in the similarity metric definition (paragraph 6.2.1 from line 424).

Response: Thanks for your comments. 
According to your suggestion, we have expanded the paragraph introducing the similarity metric definition in section 6.2.1. and included the cite to the work by Sadegh-Zadeh

Reviewer 3 Report

Very interesting and well-written article.

Author Response

Description of changes performed to meet reviewers’ requests to the manuscript "Instance-based learning following physician reasoning for assistance during medical consultation", Applied Sciences, 2021

Dear Editor-in-Chief and reviewers,

Thank you for handling/reviewing our submission. We have modified our manuscript according to the comments made by the reviewers and we have prepared a detailed response to each issue pointed out by each reviewer. We have submitted the updated manuscript, where new content is marked in blue font. The answers to reviewers are provided in this letter.

We improved our article taking into account all the comments made by the reviewers. The main changes in the manuscript included:

  • The abstract was re-written to better describe the content of the article and the quantitative results.
  • We included a comparison with a Bayesian learning approach, such as the one applied in previous works.
  • We have included a new section to present specific comments about the usability of the proposed method.
  • We have included proper references and further explained some important concepts about the proposal.

In our opinion, by following the suggestions made by the reviewers, the structure, readability, and scientific contribution of the manuscript has been greatly improved.

The reviewers comments are copied below and our answers are detailed next. New content has been marked in blue font in the revised version of the article.

Reviewer #3

Comments:

Very interesting and well-written article.

Response: Thanks for your comment.

Round 2

Reviewer 1 Report

This version of the manuscript has been significantly improved and the authors have answered most asked questions. I recommend the acceptance of the paper for publication.